# Prevalence of Polypharmacy of Older People in a Large Brazilian Urban Center and its Associated Factors

**DOI:** 10.3390/ijerph20095730

**Published:** 2023-05-05

**Authors:** Luciano Magalhães Vitorino, Jorge Henrique Lopes Mendes, Gerson de Souza Santos, Cláudia Oliveira, Helena José, Luís Sousa

**Affiliations:** 1Department of Medicine, Faculty of Medicine of Itajubá, Itajubá 37502-138, MG, Brazil; 2Department of Medicine, Centro Universitário Ages, Paripiranga 37550-030, BA, Brazil; 3School of Health Atlântica (ESSATLA), 2730-036 Oeiras, Portugal; 4Health Sciences Research Unit: Nursing (UICISA: E), Coimbra Nursing School, 3045-043 Coimbra, Portugal; 5Comprehensive Health Research Centre (CHRC), 7000-811 Evora, Portugal

**Keywords:** polypharmacy, aged, frail elderly

## Abstract

Background: With the aging population comes greater risks associated with polypharmacy, a significant public health problem. Objective: This study aimed to identify the prevalence of polypharmacy and its associated factors through Comprehensive Geriatric Assessment (CGA) among older adults treated in primary health care (PHC) in a large Brazilian urban center. Methods: We conducted a cross-sectional study with a random sampling of 400 older adults using primary health care. Polypharmacy was defined as the cumulative use of five or more daily medications. An assessment of a sociodemographic and health survey, fear of falling, and physical disabilities affecting activities of daily living and instrumental activities of daily living was conducted. Results: The mean age was 75.23 (SD: 8.53) years. The prevalence of polypharmacy and hyperpolypharmacy was 37% (*n* = 148) and 1% (*n* = 4), respectively. The adjusted logistic regression showed that participants with chronic non-communicable diseases (CNCDs) (OR = 9.24; *p* = 0.003), diabetes (OR = 1.93; *p* = 0.003), and obesity (OR = 2.15; *p* = 0.005) were associated with a greater propensity to use polypharmacy. Conclusion: Our results show that older adults with CNCDs, diabetes, and obesity were more likely to use polypharmacy. The results reinforce the importance of using CGA in clinical practice in PHC.

## 1. Introduction

The world’s population is aging rapidly, especially in developing countries such as Brazil [1]. With the aging of the population comes a greater risk of chronic non-communicable diseases (CNCDs) and greater demand for medications such as antihypertensives, lipid modifier agents, antidiabetics, non-steroidal anti-inflammatory drugs, and antidepressants [2]. In some situations, there is greater exposure to the use of polypharmacy, which is a significant public health problem due to its consequences on the older population, their families, and the public health system [3].

The indiscriminate use of polypharmacy is a significant public health challenge [4]. There is no consensus on the concept of polypharmacy. In the present study, we followed the criteria most used in the international literature, which is the use of five or more medications per day [4]. The international prevalence of polypharmacy is heterogeneous, ranging from 10% to 90% [5]. A study in 17 European countries and Israel identified a prevalence between 26.3% and 39.9% of polypharmacy use among people aged 65 and over [6]. Among older adults in developing countries such as China and Ethiopia, the prevalence is 48% and 33%, respectively. In Brazil, another developing country, data on polypharmacy are also heterogeneous [7,8]. A cross-sectional study with 9412 people aged 50 years or older identified a prevalence of 13.5% [7]. Another study with 6844 people aged 60 years or older identified a prevalence of 18% [8]. The main factors associated with the use of polypharmacy among the Brazilian older population were over 75 years of age, low education, worse perception of health status, obesity, previous history of hospitalization, diabetes, and systemic arterial hypertension [7,8].

Several studies reinforce that the use of polypharmacy is associated with worse physical and mental health outcomes and increased mortality [3,9,10]. A systematic review with more than 90,000 participants from North America, Europe, Asia, and Australia identified conflicting evidence on adverse drug reactions and effects from polypharmacy [3]. However, there was a consensus on the relationship between polypharmacy and increased demand for health care, such as hospitalization and unplanned admissions [10]. A meta-analysis of 24 studies (*n* = 2,967,952) identified that polypharmacy (≥5 drugs) increases the risk of death by 1.28 times, and hyper polypharmacy (≥10 drugs) increases the risk of death by 1.44 times [10]. Information such as increased risk of hospitalization (RR: 1.50) was also explored in the study. Other evidence shows that polypharmacy increases the risk of urinary incontinence [11], falls [12], dementia [13], and diabetes in the older population [14].

According to the Pan American Health Organization, active aging should be able to maintain and improve health, and the indiscriminate use of polypharmacy can go against the concept of healthy aging as there is no maintenance of quality of life [15,16]. In addition to higher public spending on health, the inadequate prescription of medication is costly to the public health system, both for the acquisition of the medication and for the possible negative outcomes, which demand even more expenses. It is estimated that in the United States, for every 1 dollar in medication, $1.33 is spent on treating adverse medication effects. At the global level, poorly managed polypharmacy contributed to 4% of total avoidable health costs, with a total cost of US $18 billion, in other words, 0.3% of health expenditure [15].

Health professionals and researchers are very interested in exploring the impacts of polypharmacy on the health and quality of life of the older population. In 2019, polypharmacy was included as one of the three main areas of action of the third Global Patient Safety Challenge of the World Health Organization [16]. In this regard, it is imperative to investigate the risks and protective factors for the use of polypharmacy in the older populations of developing countries, with 31.2 million (17.7%) people aged 60 or over and with major health challenges in the Brazilian health system [17]. Although other studies have addressed polypharmacy among institutionalized and hospitalized older people, the topic still needs to be explored among the older Brazilian population assisted in primary health care (PHC). Another important aspect is that many studies have investigated the association of polypharmacy with two or three health outcomes [18,19]. Due to the complexity and possible impacts on the health of older adults, it is crucial to verify the effects of polypharmacy using a Comprehensive Geriatric Assessment (CGA) [20]. CGA in clinical practice in PHC positively impacts the treatment and quality of life of older adults [20]. In order to fill some gaps in the literature, the objective of this study was to identify the prevalence of polypharmacy and its associated factors through CGA (variables sociodemographic characteristics, history of falls, fear of falling, activities of daily living, CNCDs, depressive symptoms, and cognitive function) among older adults treated in primary health care in a large Brazilian urban center.

## 2. Materials and Methods

### 2.1. Study Design

This study is part of a longitudinal multidimensional research project with older adults in the city of São Paulo, SP, Brazil [21]. This is a cross-sectional study, with a probabilistic sampling of older adults enrolled in a Basic Health Unit (BHU) in the eastern region of the city of São Paulo, SP. The Ethics and Research Committee of the Municipal Health Department of São Paulo, under protocol #2.364.869, approved this project. All participants were informed about the objectives of the study and signed the Informed Consent Form.

### 2.2. Location, Population, and Sample

The study was carried out at the Marcus Wolosker (Belenzinho) BHU by one of the authors, who has a Ph.D. in Collective Health and 10 years of experience as a nurse in primary health care (PHC). In 2018, Belenzinho BHU had around 5000 people aged 60 and over. We use the G*Power program 3.1.9.7 (Heinrich-Heine-University, Düsseldorf, Germany) to calculate the statistical power of the analyses. Aiming to perform multivariate logistic regression models, with 11 independent variables (variables with *p* < 0.20 in the bivariate logistic regression), 13.5% prevalence of polypharmacy [7], two-tailed *p*-value, α = 0.05, and 400 participants. The post hoc analysis showed a statistical power of 98%.

### 2.3. Data Collection, Inclusion, and Exclusion Criteria

Data were collected between November 2017 and August 2018. We used medical record numbers to select potential participants for the survey. Those chosen were informed about the research during the nursing consultations at the BHU. The interviews (lasting approximately 40 min) were carried out individually in a private environment during the nursing consultation. People aged 60 or over and registered at Belenzinho BHU participated in the survey. Participants with a medical diagnosis of dementia, Alzheimer’s, or severe cognitive or physical impairment did not participate in the research.

### 2.4. Dependent Variable

Although there is no consensus on the concept of polypharmacy (numerical counts or numerical counts and duration), we adopted the most used concept among health professionals and researchers, which is the use of 5 medications or more per day [4]. The following question was asked, “How many medications do you take per day?”. In case the participant did not remember or was not sure, the researcher checked the number of medications in the participant’s medical record. The results were transformed into categories. Polypharmacy (≥5 medications) yes or no; or hyperpolypharmacy (≥10 medications) yes or no.

### 2.5. Independent Variables

Regarding sociodemographic variables, the following were evaluated: age group (60–70 years; 70–79; 80 or more); gender (male or female); marital status (with or without a partner); knowing how to read or write (yes or no); years of study (none, 1 to 4 years, 5 or more); lives alone (yes or no); satisfaction with life (yes or no); health perception (poor, regular, good or excellent); chronic illness (we consider that chronic illness is persistent health conditions typically have a prolonged course and stem from a multifaceted interplay of genetic, physiological, environmental, and behavioral elements) (yes or no); daily medication use (yes or no); smoking (yes or no) and history of falls (yes or no); last fall < 12 months (yes or no); basic activities of daily living [22] (dependent or independent); instrumental activities of daily living [22] (dependent or independent); and fear of falling [23] (afraid and not afraid).

Comorbidities [21]: obesity (yes or no); smoking (yes or no); hypertension (yes or no); diabetes (yes or no); cardiovascular disease (yes or no); neoplasm (yes or no); lung disease (yes or no); disease of the musculoskeletal system (yes or no); neurological (yes or no); and cardiometabolic (yes or no). This information was collected from the participant’s medical records.

Fear of falling: fear of falling was assessed using the Falls Effectiveness Scale (FES-I), developed in 2005 by Yardley and validated for Portuguese in 2010 [23]. This scale assesses fear of falling using 16 areas of different activities through a Likert scale of 4 points in each item, totaling a minimum of 16 points and a maximum of 64. With regards to scoring, 16 to 22 points were adopted as low concern and 23 to 64 points as high concern [24].

Basic activities of daily living (BADL): Assessed using the KATZ index—a tool developed in 1976 by Sidney Katz. It was validated for Portuguese in 2008 and used the degree of independence in the result of six functions [22]. The results were scored in a range of 0 to 6 points, classifying the individual as independent (zero points) or dependent (1 or more points).

Instrumental activities of daily living (IADL): Lawton Scale—developed by Lawton and Brody in 1969. It was validated for Portuguese in 2008 and was used to assess the instrumental activities of daily living (IADL) [25]. It is based on nine items that allow a total score of 0 to 21 points, being classified as an independent individual (21 points) or dependent (20 or less points).

### 2.6. Data Analysis

Data were analyzed using the Statistical Package for Social Sciences—SPSS 26 program (IBM Corp., Armonk, NY, USA). A descriptive analysis (absolute frequency and relative frequency) was performed to describe the sociodemographic and health characteristics of the participants.

Hypothesis tests were performed as follows. First, a series of unadjusted logistic regression models were performed using the polypharmacy outcome (0 = no polypharmacy or 1 = with polypharmacy) and the independent variables. Second, the variables that presented *p* < 0.10 in the unadjusted logistic regression analysis were included in the multivariate logistic regression model (health perception *p* = 0.0035; chronic disease *p* = 0.002; disease of the musculoskeletal system *p* = 0.029; hypertension *p* = 0.010; diabetes *p* = 0.001; and obesity *p* = 0.017). The logistic regression model was run using the “stepwise forward selection” technique. A significance level of 5% was chosen for the test, with a 95% confidence interval.

## 3. Results

A total of 488 participants were invited to participate in the survey, of which, 400 (83.33%) completed all items in the questionnaire. Table 1 shows the comparison of participants who used and who did not use polypharmacy. The mean age was 75.23 (SD: 8.53) years. The majority were female (63.2%), without a partner (67%), 40% could not read and write, and about 1/3 of the older adults lived alone. Dissatisfaction with life was reported by most participants (54.25%). As for self-perception of health status, 62.6% of older adults rated it as poor or fair, and 92.3% had at least one CNCD. About 62.70% of the participants had fallen, 24% had fallen in the last 12 months, and 90.50% had FOF. About 9 out of 10 older adults claimed to use medication daily, using an average of 3.85 (SD: 2.01) medications per day. The prevalence of polypharmacy (≥5 per day) use was 37% (*n* = 148), and of hyperpolypharmacy (≥10 per day) was 1% (*n* = 4). We identified that participants with a chronic illness were more likely to use polypharmacy (*p* < 0.001). We did not identify a significant difference in the other comparisons (*p* > 0.05).

The description of CNCDs by systems is shown in Table 2. Most had cardiovascular disease (73.70%) and 4.7% had some type of neoplasm. We identified that participants with musculoskeletal disorders were significantly less likely to engage in polypharmacy (*p* = 0.028).

Table 3 shows the results of the unadjusted logistic regression. Participants with good health perception (OR = 1.84; *p* = 0.035), with CNCDs (OR = 9.50; *p* = 0.002), with musculoskeletal disease (OR = 1.67; *p* = 0.029), hypertension (OR = 1.81; *p* = 0.01), diabetes (OR = 2.06; *p* = 0.001), and obesity (OR = 1.98; *p* = 0.017) were associated with a greater propensity to use polypharmacy.

Table 4, adjusted logistic regression, showed that participants with CNCDs (OR = 9.24; *p* = 0.003), diabetes (OR = 1.93; *p* = 0.003), and obesity (OR = 2.26; *p* = 0.005) were associated with a greater propensity to use polypharmacy.

## 4. Discussion

This study investigated the prevalence of polypharmacy among older adults assisted in PHC and, through a CGA, verified the factors associated with polypharmacy. Although the prevalence of polypharmacy found in the literature is heterogeneous, our results were similar to those in the literature compared with Brazilian and international evidence [7,8,10,12]. Our results show that older adults with CNCDs, diabetes, and obesity were more likely to use polypharmacy. These results reinforce the importance of investigating the use of polypharmacy and highlight that having CNCDs (i.e., diabetes or high blood pressure) is among the risk factors. Communication between health professionals, especially physicians, pharmacists, and the targeted review of medications is an important part of managing care for chronic conditions in the older population [27,28].

The prevalence of polypharmacy in the present study is in line with previous evidence [6,29]. Depending on the age group, the concept of polypharmacy, the health context of the population, and the conditions of access to health, polypharmacy can vary widely [29,30]. Evidence reinforces that the sociodemographic and clinical profile of the participants in our research (i.e., older people with CNCDs and dependents) is strongly associated with a greater risk of indiscriminate use of polypharmacy [3,17]. Contrasting our results, a study with a representative sample of people aged 50 or over from 70 municipalities with a clinical profile similar to that of our study (i.e., health problems, CNCDs, poor health status, and obesity) identified a prevalence of almost three times lower [7]. A possible explanation for this amplitude is that the participants in our study are users of a BHU, and there is a close relationship between greater access to health care and polypharmacy [7]. The place where the research participants resided was of low economic power and these findings were also found in research carried out in Italy and England [6,31].

We identified that the presence of CNCD was strongly associated with a greater propensity for polypharmacy. Several studies reinforce this relationship [6,18,32]. A study carried out with 3904 older adults from five European countries identified that having a chronic disease was associated with the risk of polypharmacy, and participants diagnosed with eight diseases or more showed a significant increase [32]. These results are noteworthy because there are estimates that approximately half of the people aged 65 years or older have at least three CNCDs and 20% have five or more [33]. There is no doubt that the aging of the population and the presence of CNCDs are strongly associated with a higher risk of polypharmacy [29,34]. For this reason, health professionals must be aware of the inappropriate prescription of medication to the older population.

Polypharmacy in people with diabetes is associated with increased all-cause mortality, macrovascular complications, and hospitalization [14]. With regard to hypertension and cardiometabolic risk, a study involving 7621 people with cardiometabolic risk found that polypharmacy was observed in 29.7% of hypertensive patients and in 82.4% of people who had the three cardiometabolic risk factors [35]. Polypharmacy in hypertensive people is associated with an increased risk of adverse events, non-adherence to medication, drug interactions, drug–food interactions, and poor management of the drug regimen [36,37]. Knowledge of drugs and foods that interact with known antihypertensive agents is mandatory, as well as the fact that innovative approaches must be implemented for drug regimen management. Medication reconciliation is one of these approaches and could contribute to reducing the use of polypharmacy. Medication reconciliation is defined as a formal process for enabling the creation of the most complete and accurate list possible of a patient’s current medications and allowing the comparison of this list with the patient’s records or even with new prescriptions [38].

This study has some limitations that should be identified. First, the research was based on self-reported information based on memories, which may imply information bias. Second, the definition of polypharmacy was based on a drug cohort of ≥5 therapeutic classes; however, participants may be taking non-prescription medications. Third, the prevalence of polypharmacy estimated in our study may be underestimated and may be much higher. Fourth, the use of ORs from logistic regression may have overestimated the associations, and given the high prevalence of polypharmacy, we recommend future studies of these types of methods that can estimate prevalence ratios, such as log-binomial modeling or Poisson regression. Finally, this study was carried out with users of a BHU. Thus, the findings may be generalizable only to BHU users.

Finally, this study has clinical implications that should be highlighted. The results of our study reinforce the need for greater surveillance of the use of prescribed medications and multidisciplinary interventions that optimize the balance between benefits and harm in the prescription of medications [28]. Clinical staff should review the medication regimen (which is different from medication reconciliation), and use screening tools, pharmacist-led interventions, and computer-based strategies [28,39]. Medication reconciliation can decrease therapeutic errors and adverse effects, improve patient safety, and decrease negative outcomes in older people at higher risk [28,39].

## 5. Conclusions

This study shows that the prevalence of polypharmacy is high among older people in our study. Our results show that older adults with CNCDs, diabetes, and obesity were more likely to use polypharmacy. These results reinforce the importance of using Comprehensive Geriatric Assessment in clinical practice in PHC.

## Figures and Tables

**Table 1 ijerph-20-05730-t001:** Sociodemographic and health comparison of participants with and with no use of polypharmacy (*n* = 400) [21].

Variables	Total	Polypharmacy		*p*-Value
		Yes	No	
	*n* (%)	*n* (%)	*n* (%)	
Age				
60–69	104 (26.00)	39 (26.30)	65 (25.80)	0.687
70–79	159 (39.70)	55 (37.20)	104 (41.30)	
≥80 years	137 (34.30)	54 (36.50)	83 (32.90)	
Gender				
Male	147 (36.80)	47 (31.80)	152 (60.30)	0.112
Female	253 (63.20)	101 (68.20)	100 (39.70)	
Marital status				
With partner	132 (33.00)	46 (31.10)	86 (34.10)	0.532
Without Partner	262 (67.00)	102 (68.9)	166 (65.90)	
Knows how to read/write				
Yes	242 (60.50)	51 (34.50)	145 (57.50)	0.114
No	158 (39.50)	97 (65.50)	107 (42.50)	
Lives alone				
Yes	126 (31.50)	46 (31.10)	80 (31.70)	0.890
No	274 (68.50)	102 (68.90)	172 (68.30)	
Satisfied with life				
Yes	183 (45.80)	64 (43.20)	119 (47.20)	
No	217 (54.20)	84 (56.80)	133 (52.80)	0.441
Health perception				
Poor	89 (22.30)	41 (27.70)	48 (19.00)	
Regular	161 (40.30)	60 (40.50)	101 (40.20)	
Good	120 (30.00)	38 (25.70)	82 (32.50)	0.155
Excellent	30 (7.40)	9 (6.10)	21 (8.30)	
Chronic illness				
Yes	369 (92.30)	146 (98.60)	223 (88.50)	<0.001
No	31 (7.70)	2 (1.40)	29 (11.50)	
Smoker				
Yes	110 (27.50)	42 (28.40)	68 (27.00)	0.763
No	290 (72.50)	106 (71.60)	184 (73.00)	
History of falls ^a^				
Yes	251 (62.70)	97 (66.50)	155 (61.50)	0.124
No	149 (37.30)	51 (34.50)	97 (38.50)	
Last fall (<12 months)				
Yes	96 (24.00)	112 (76.00)	206 (81.90)	0.259
No	304 (76.00)	36 (24.00)	46 (18.10)	
Katz				
Dependent	108 (27.00)	42 (28.40)	66 (26.20)	0.634
Independent	292 (73.00)	106 (71.60)	186 (73.80)	
Lawton				
Dependent	157 (39.35)	60 (40.50)	98 (38.90)	0.744
Independent	242 (60.65)	88 (59.50)	154 (61.10)	
FES-I ^b^				
With fear	362 (90.50)	137 (92.60)	225 (89.30)	0.280
Without fear	38 (9.50)	11 (7.40)	27 (10.70)	
Use of medication ^c^				
Yes	363 (90.70)	-	-	
No	37 (9.30)			
Polypharmacy (≥5 per day)		-	-	
Yes	148 (37.00)			
No	252 (63.00)			
Hyperpolypharmacy (≥10 per day)				
Yes	04 (1.00)	-	-	
No	396 (99.00)			

a. History of falls in the last 12 months. b. Adopted 16 to 22 points for without fear and 23 to 64 points for with fear. c. Daily; FES-I: Falls Efficacy Scale International.

**Table 2 ijerph-20-05730-t002:** Prevalence of non-communicable chronic diseases (*n* = 400) [21].

Comorbidity	Total	Polypharmacy		*p*-Value
		Yes	No	
	*n* (%)	*n* (%)	*n* (%)	
Cardiovascular				
Yes	295 (73.70)	116 (78.40)	179 (71.00)	0.107
No	105 (26.30)	32 (21.60)	73 (29.00)	
Neoplasm				
Yes	19 (4.70)	4 (2.70)	15 (6.00)	0.140
No	381 (95.30)	144 (97.30)	237 (94.00)	
Pulmonary				
Yes	26 (6.50)	6 (4.10)	20 (7.90)	0.128
No	374 (93.50)	142 (95.90)	232 (92.10)	
Musculoskeletal				
Yes	118 (29.50)	34 (23.00)	84 (33.30)	0.028
No	282 (70.50)	114 (77.00)	168 (66.70)	
Neurological				
Yes	199 (49.80)	69 (46.60)	130 (51.60)	0.338
No	201 (50.20)	79 (53.40)	122 (48.40)	
Cardiometabolic				
Yes	206 (51.50)	83 (56.10)	123 (48.80)	0.160
No	194 (48.50)	65 (43.90)	129 (51.20)	

**Table 3 ijerph-20-05730-t003:** Binary logistic regression between polypharmacy and multidimensional assessment of participants (*n* = 400).

		Polypharmacy ^a^	
	β (E.P)	OR Non-Adjusted (CI 95%)	*p*-Value
Age (average)	0.003 (0.012)	0.997 (0.97–1.02)	0.997
Gender (female)	0.346 (0.219)	1.414 (0.92–2.17)	0.113
Read/write (yes)	0.339 (0.215)	1.404 (0.92–2.13)	0.115
Marital status (single)	0.139 (0.222)	1.149 (0.74–1.74)	0.532
Family arrangement (without partner)	0.031 (0.223)	0.970 (0.62–1.50)	0.890
Health perception (regular)	0.363 (0.268)	1.438 (0.85–2.43)	0.175
Health perception (good)	0.612 (0.289)	1.843 (1.04–3.25)	0.035
Health perception (very good)	0.690 (0.452)	1.993 (0.82–4.83)	0.127
Chronic disease (yes)	2.251 (0.739)	9.500 (2.23–40.39)	0.002
Musculoskeletal (yes)	0.517 (0.237)	1.676 (1.05–2.66)	0.029
Hypertension (yes)	0.596 (0.230)	1.810 (1.15–2.85)	0.010
Diabetes (yes)	0.727 (0.216)	2.060 (1.35–3.15)	0.001
Obesity (yes)	0.685 (0.288)	1.980 (1.12–3.49)	0.017
Smoking (yes)	0.070 (0.231)	1.072 (0.68–1.68)	0.763
Katz (dependent)	0.120 (0.232)	1.127 (0.71–1.77)	0.606
Lawton (dependent)	0.052 (0.212)	1.054 (0.69–1.59)	0.806
Depression (yes)	0.287 (0.216)	0.750 (0.49–1.14)	0.184
Cognition impairment (yes)	0.032 (0.104)	1.003 (0.93–1.26)	0.098
Fear of falling (With fear)	0.400 (0.374)	1.500 (0.71–3.10)	0.282
Falls in the last 12 months (yes)	0.357 (0.317)	1.429 (0.76–2.66)	0.261

a. Polypharmacy 1 = yes; 0 = no; CI: Confidence Interval; OR: odds ratio. MMSE: Mini-mental state exam. MMSE cutoff point for detecting cognitive impairment: <23 [26].

**Table 4 ijerph-20-05730-t004:** Multivariate logistic regression between polypharmacy and multidimensional assessment of participants (*n* = 400).

		**Polypharmacy ^a^**	
	**β (E.P)**	**Adjusted OR (CI 95%)**	** *p* ** **-Value**
Chronic disease (yes)	2.224 (0.743)	9.249 (2.15–39.70)	0.003
Diabetes (yes)	0.661 (0.222)	1.936 (1.25–2.99)	0.003
Obesity (yes)	0.817 (0.294)	2.264 (1.27–4.03)	0.005

a. Polypharmacy 1 = yes; 0 = no; S.E.: Standard Error; CI: Confidence Interval; OR: odds ratio.

## Data Availability

The data presented in this study are available on request from the corresponding author.

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
