# Peer review of "Prevalence of Polypharmacy of Older People in a Large Brazilian Urban Center and its Associated Factors"

_ijerph, 2023, doi:10.3390/ijerph20095730_

Round 1

Reviewer 1 Report

In this study, the authors evaluated the prevalence of and risk factors for polypharmacy in a small of 488 individuals surveyed in Brazil. Despite the small sample size and the existence of previous studies exploring this topic in larger cohorts of older adults in Brazil, the authors add some novelty to existence literature by including evaluation of comprehensive geriatric assessment and conducting a more rigorous analysis of risk factors associated with PP. However, to overcome the flaws of small sample size and cross-sectional design of the study, a more detailed description of study cohort (especially regarding the qualitative component of pharmacological history) should be provided, as well as some refinements of statistical analysis. I have some concerns that I report here below:

1) Statistical analysis: in logistic regression for frequent outcomes (as PP is in this study), prevalence rate ratio (PRRs) use is encouraged in the place of OR, as OR often leads to overestimation or misinterpretation of associated measures (see https://www.ncbi.nlm.nih.gov/pmc/articles/PMC5135596).

2) Dependent variable: to ensure a more complete evaluation of surveyed patients, I suggest to include the number of patients with hyperpolypharmacy in both study abstract and study results. Add definition of hyperpolypharmacy (>= 10 medications). Additionally, are there any risk factors associated specifically with hyperpolypharmacy?

3) Independent variables: which is the definition of chronic illness? Additionally, functional variables should be reported as all continuous or all categorical. Use a cut-off for defining cognitive impairment based on MMSE.

4) A qualitative description of medications should be added to the present study. Which are the main classes of medications that contributed to the polypharmacy risk in this population? Which are the patterns of medications used by patients with polypharmacy? Are there any peculiar associations between different classes?

5) Table 1 and 2: were patients with polypharmacy characterized by a different demographic and clinical profile compared to those with no polypharmacy? I suggest to include stratification by polypharmacy to table 1 and add a between-group to test for assessment of statistical significance between these two groups.

6) Logistic regression results: chronic disease is strongly associated with PP. This is somewhat obvious and collinear with study outcome, as medications are used for diseases. Additionally, >90 % of the study population has chronic disease, so leading to very wide confidence intervals. For these reasons, I suggest to add a further adjusted logistic regression model excluding the variable chronic disease to the presented one (in order to evaluate if other risk factors emerged as significant ones after adjustment).

-Discussion: many functional factors of the CGA (BADL and cognitive function) were not associated with polypharmacy. A brief debatement on not found associations (study results vs previous literature) as well as speculated reasons behind not found associations should be added to discussion. I suggest to make it more "comprehensive", adding these elements to more classical findings of association with chronic disease and single diseases.

Author Response

Letter in response to the reviewers’ comment

April 17th, 2023

Dear Editor

Thank you for reviewing the manuscript entitled " Prevalence of Polypharmacy of Older People in a Large Brazilian Urban Center and its Associated Factors". We have now addressed the reviewers’ comments. We hope our manuscript could be published now.

Best regards,

The Authors

Reviewer 1

First, we wish to take this opportunity to thank you for the valuable and constructive comments, so that we can see the shortcomings of the paper. These comments are all valuable for improving our paper. All the comments have been carefully revised and highlighted.

In this study, the authors evaluated the prevalence of and risk factors for polypharmacy in a small of 488 individuals surveyed in Brazil. Despite the small sample size and the existence of previous studies exploring this topic in larger cohorts of older adults in Brazil, the authors add some novelty to existence literature by including evaluation of comprehensive geriatric assessment and conducting a more rigorous analysis of risk factors associated with PP. However, to overcome the flaws of small sample size and cross-sectional design of the study, a more detailed description of study cohort (especially regarding the qualitative component of pharmacological history) should be provided, as well as some refinements of statistical analysis.

Response: Thank you very much for your recognition of us.

I have some concerns that I report here below:

Statistical analysis: in logistic regression for frequent outcomes (as PP is in this study), prevalence rate ratio (PRRs) use is encouraged in the place of OR, as OR often leads to overestimation or misinterpretation of associated measures (see https://www.ncbi.nlm.nih.gov/pmc/articles/PMC5135596).

Response: A major objective of this study was to identify the prevalence of Polypharmacy and its associated factors through Comprehensive Geriatric Assessment (CGA) among older adults treated in primary health care (PHC) in a large Brazilian urban center. Logistic regression has been used in other studies such as the present one to examine associations of this type, so there is a precedent for using the present analytic approach (e.g., Ye L, Yang-Huang J, Franse CB, Rukavina T, Vasiljev V, Mattace-Raso F, Verma A, Borrás TA, Rentoumis T, Raat H. Factors associated with polypharmacy and the high risk of medication-related problems among older community-dwelling adults in European countries: a longitudinal study. BMC Geriatr. 2022 Nov 7;22(1):841. doi: 10.1186/s12877-022-03536-z. PMID: 36344918; PMCID: PMC9641844. ; Rieckert A, Trampisch US, Klaaßen-Mielke R, Drewelow E, Esmail A, Johansson T, Keller S, Kunnamo I, Löffler C, Mäkinen J, Piccoliori G, Vögele A, Sönnichsen A. Polypharmacy in older patients with chronic diseases: a cross-sectional analysis of factors associated with excessive polypharmacy. BMC Fam Pract. 2018 Jul 18;19(1):113. doi: 10.1186/s12875-018-0795-5. PMID: 30021528; PMCID: PMC6052592. ; Turner JP, Shakib S, Singhal N, Hogan-Doran J, Prowse R, Johns S, Bell JS. Prevalence and factors associated with polypharmacy in older people with cancer. Support Care Cancer. 2014 Jul;22(7):1727-34. doi: 10.1007/s00520-014-2171-x. Epub 2014 Mar 2. Erratum in: Support Care Cancer. 2014 Jul;22(7):1735. PMID: 24584682.).  Therefore, in response to the Reviewer’s concern, in 2021, a Brazilian paper was published that aimed to evaluate the prevalence and factors associated with polypharmacy in older adults from a rural area. The authors used OR to assess the association. The prevalence of polypharmacy was 40%, close to that of the present study. Still, though, we acknowledge the Reviewer’s concern above as a study limitation in the Limitation’s section part of the Discussion.

2) Dependent variable: to ensure a more complete evaluation of surveyed patients, I suggest to include the number of patients with hyperpolypharmacy in both study abstract and study results. Add definition of hyperpolypharmacy (>= 10 medications). Additionally, are there any risk factors associated specifically with hyperpolypharmacy?

Response: Thanks for the recommendation. We have added the hyperpolypharmacy prevalence (1%; n=04). The percentage of hyperpolypharmacy was low, for this reason we maintained the association analyzes for polypharmacy.

3) Independent variables: which is the definition of chronic illness? Additionally, functional variables should be reported as all continuous or all categorical. Use a cut-off for defining cognitive impairment based on MMSE.

Response: Thanks for the recommendations. We have added the chronic illness’ definition and MMSE cutoff point for detecting cognitive impairment: <23

4) A qualitative description of medications should be added to the present study. Which are the main classes of medications that contributed to the polypharmacy risk in this population? Which are the patterns of medications used by patients with polypharmacy? Are there any peculiar associations between different classes?

Response: We understand your concern. If it were possible to add it would be very important for the article. Unfortunately, we do not have this data in our research project.

5) Table 1 and 2: were patients with polypharmacy characterized by a different demographic and clinical profile compared to those with no polypharmacy? I suggest to include stratification by polypharmacy to table 1 and add a between-group to test for assessment of statistical significance between these two groups.

Response: Thanks for the suggestions. We have added the difference between groups to table 1 and table 2. We believe it is now more informative

6) Logistic regression results: chronic disease is strongly associated with PP. This is somewhat obvious and collinear with study outcome, as medications are used for diseases. Additionally, >90 % of the study population has chronic disease, so leading to very wide confidence intervals. For these reasons, I suggest to add a further adjusted logistic regression model excluding the variable chronic disease to the presented one (in order to evaluate if other risk factors emerged as significant ones after adjustment).

Response: We understand your concern. We performed the suggested analyzes, but there was no change (new variables), only a change in the effect size of the variable diabetes and obesity. For this reason, we opted to leave the analyzes as they are. See the output:

-Discussion: many functional factors of the CGA (BADL and cognitive function) were not associated with polypharmacy. A brief debatement on not found associations (study results vs previous literature) as well as speculated reasons behind not found associations should be added to discussion. I suggest to make it more "comprehensive", adding these elements to more classical findings of association with chronic disease and single diseases.

Response: We appreciate the comment. Our study did not show an association between polypharmacy, as was found in previous studies (Chuang et al., 2023; Gutiérrez-Valencia et al, 2019). Excessive polypharmacy was associated with worse BADL performance, higher level of frailty and higher prevalence of diabetes mellitus and chronic kidney disease (Chuang et al., 2023). Another study found that the factors most associated with polypharmacy were the number of chronic diseases and the degree of dependence on BADLs (Gutiérrez-Valencia et al, 2019).

Chuang, Y. N., Chen, C. C., Wang, C. J., Chang, Y. S., & Liu, Y. H. (2023). Frailty and polypharmacy in the community‐dwelling elderly with multiple chronic diseases. Psychogeriatrics23(2), 337-344. https://doi.org/10.1111/psyg.12936

Gutiérrez-Valencia, M., Herce, P. A., Lacalle-Fabo, E., Escámez, B. C., Cedeno-Veloz, B., & Martínez-Velilla, N. (2019). Prevalence of polypharmacy and associated factors in older adults in Spain: Data from the National Health Survey 2017. Medicina Clínica (English Edition)153(4), 141-150. https://doi.org/10.1016/j.medcle.2019.06.009

Reviewer 2 Report

Tables are not clearly described. P value of 0.2 is not standard for such studies. It is typical to use p<0.05 or 0.1. The alpha is set at 0.05 so p-value less than 0.05 only should be reported as significant. The paper could benefit from a review by a statistician. The issue of polypharmacy is described in detail but the study does not have any novel findings other than saying polypharmacy is a concern in the study population. Why it is a concern, which medications are potentially concerning, are there medications that elderly in the study should not be on (Beer's criteria) - these are all unanswered questions.

Author Response

Letter in response to the reviewers’ comment

April 17th, 2023

Dear Editor

Thank you for reviewing the manuscript entitled " Prevalence of Polypharmacy of Older People in a Large Brazilian Urban Center and its Associated Factors". We have now addressed the reviewers’ comments. We hope our manuscript could be published now.

Best regards,

The Authors

Reviewer 2

Dear Reviewer, 2

Thanks for all comments. This opportunity that you have given us allows us to improve the quality of our work. All questions were carefully analyzed, and we tried to respond fully to what was requested.

  1. Tables are not clearly described.

Response: We describe the comparison that was statistically significant. We also removed some descriptions, noted that it was redundant. I hope it's clearer.

  1. P value of 0.2 is not standard for such studies. It is typical to use p<0.05 or 0.1. The alpha is set at 0.05 so p-value less than 0.05 only should be reported as significant. The paper could benefit from a review by a statistician.

Response: As suggested, we used p<.10 for the logistic regression models. Only the OR for obesity had a slight change.

  1. The issue of polypharmacy is described in detail but the study does not have any novel findings other than saying polypharmacy is a concern in the study population. Why it is a concern, which medications are potentially concerning, are there medications that elderly in the study should not be on (Beer's criteria) - these are all unanswered questions.

Response: Thanks for the critical analysis. Indeed, if meticulous classes were used, new evidence could emerge. As previously reported, unfortunately there is no data on drug classes. However, I would like to make a counterpoint. The present study may reinforce the importance of the Comprehensive Geriatric Assessment in the conduct of care for elderly people, and in the present study the use of polypharmacy.

Round 2

Reviewer 1 Report

The authors addressed all my concerns.